# The Feasibility of Cervical Elastography in Predicting Preterm Delivery in Singleton Pregnancy with Short Cervix Following Progesterone Treatment

**DOI:** 10.3390/ijerph18042026

**Published:** 2021-02-19

**Authors:** Yun Ji Jung, Hayan Kwon, Jeongeun Shin, Yejin Park, Seok-Jae Heo, Hyun Soo Park, Soo-young Oh, Ji-Hee Sung, Hyun-Joo Seol, Hyun Mi Kim, Won Joon Seong, Han Sung Hwang, Inkyung Jung, Ja-Young Kwon

**Affiliations:** 1Division of Maternal-Fetal Medicine, Department of Obstetrics and Gynecology, Institute of Women’s Medical Life Science, Yonsei University College of Medicine, Yonsei University Health System, Seoul 03722, Korea; ccstty@yuhs.ac (Y.J.J.); whitekwon@yuhs.ac (H.K.); jeshin84@yuhs.ac (J.S.); drsweetrain@gmail.com (Y.P.); 2Division of Biostatistics, Department of Biomedical Systems Informatics, Yonsei University College of Medicine, Seoul 03722, Korea; sjheo@yuhs.ac; 3Department of Obstetrics and Gynecology, Dongguk University Ilsan Hospital, Dongguk University, Goyang 10326, Korea; hsparkmd@gmail.com; 4Samsung Medical Center, Department of Obstetrics and Gynecology, School of Medicine, Sungkyunkwan University, Seoul 06351, Korea; ohsymd@skku.edu (S.-y.O.); obgysung@gmail.com (J.-H.S.); 5Department of Obstetrics and Gynecology, Kyung Hee University Hospital at Gangdong, School of Medicine, Kyung Hee University, Seoul 05278, Korea; seolhj@khu.ac.kr; 6Department of Obstetrics and Gynecology, School of Medicine, Kyungpook National University Hospital, Daegu 41404, Korea; hyunmik@gmail.com (H.M.K.); wjseong@knu.ac.kr (W.J.S.); 7Department of Obstetrics and Gynecology, School of Medical, Konkuk University, Seoul 05030, Korea; hwanghs@kuh.ac.kr

**Keywords:** short cervix, elastography, progesterone, prediction, preterm delivery

## Abstract

Previous studies demonstrated an association between cervical strain and risk of spontaneous preterm delivery (sPTD). The present study aimed to assess the efficacy of elastography in predicting sPTD at <32 weeks of gestation in women with singleton pregnancies receiving progesterone for short cervix (≤2.5 cm) diagnosed between 16 and 28 weeks of gestation Among 115 participants eligible for analysis, nine had sPTD at <32 weeks. Preprogesterone (PP0) mean internal os strain (IOS), elasticity contrast index (ECI), hardness ratio (HR), one-week postprogesterone (PP1) IOS, mean external os strain (EOS), ECI, and HR were significantly different between groups. Higher PP0 IOS, PP1 IOS, and PP1 EOS were associated with a 2.92, 4.39 and 3.65-fold increase in the risk of sPTD at <32 weeks, respectively (adjusted for cervical length (CL) at diagnosis; *p* = 0.04, 0.012 and 0.026, respectively). A combination of CL at diagnosis, PP0 IOS and PP1 EOS showed a significantly higher area under the receiver operating characteristic curve (0.858) than that of CL alone (*p* = 0.041). In women with singleton pregnancies receiving progesterone for short cervix, cervical elastography performed before and one week after progesterone treatment may be useful in predicting sPTD at <32 weeks of gestation.

## 1. Introduction

The efficacy of progesterone treatment in women with short cervix (≤2.5 cm) at the mid-trimester for preventing preterm birth has been widely studied, and study results have been promising. Recently, a meta-analysis of data from women with short cervix diagnosed at the mid-trimester demonstrated that vaginal progesterone therapy significantly reduced preterm birth at <33 weeks of gestation by 38% and 31% in singleton and twin pregnancies, respectively [1,2]. Nonetheless, preterm birth occurs in one-third of women with short cervix undergoing progesterone treatment [3]. Conflicting results regarding the efficacy of progesterone in pregnant women with a short cervix have been reported depending on the involved risk factors [1,4]. Plausible reasons may be as follows: (1) cervical length (CL) alone does not yield high positive predictive value (PPV) in predicting sPTD [5,6]; (2) the mechanism of cervical shortening may be different from the target mechanism of progesterone and (3) patients who may benefit from progesterone were not carefully selected as a valid test for screening was lacking. Therefore, a clinical test that can predict whether progesterone treatment will be successful in patients with short cervix would help more selectively differentiate those who would benefit with progesterone treatment from the nonresponsive group, for which other treatment options, including cerclage, should be considered for preventing sPTD.

Among studies on potential methods that can complement CL measurement for predicting PTD, those on cervical elastography are encouraging [7,8,9,10]. Elastography is an ultrasonographic method that produces images and evaluates the stiffness of soft tissues. This technique is performed in patients with liver [11,12], thyroid [13,14] or breast diseases to assess tissue elasticity in different pathologic states such as inflammation or malignancy [15,16]. Recently, in the field of obstetrics and gynecology, the stiffness of the cervical tissue and extracellular matrix (ECM) has been evaluated using elastography. Cervical elastography assesses changes in consistency during the course of the maturation process prior to delivery, such as ripening, effacement, and dilatation. Cervical elastography provides information about cervical stiffness and helps predict preterm delivery or successful labor induction [17].

Most studies on cervical elastography have evaluated the association between having a short cervix, regardless of receiving progesterone, and preterm birth [9,10,18]. These studies demonstrated that in pregnant women with short CL, the PPV for predicting preterm birth can be enhanced by adding a cervical strain value rather than using CL alone. However, its feasibility for predicting sPTD in the population with short cervix receiving progesterone treatment is unclear. The theoretical mechanism of progesterone is hypothesized to be through the inhibition of cervical ripening by controlling cervical ECM degradation and remodeling via anti-inflammatory activity and suppression of matrix metalloproteinase inhibitors [19,20,21,22]. Thus, progesterone-driven cervical changes may be detected by measuring cervical strain in women with short cervix who respond to progesterone treatment.

Cervical elastography modalities differ according to the method of tissue assessment. Compared to other elastography modalities, E-cervix^TM^ (WS80A; Samsung Medison, Seoul, Korea) is advantageous in that it is relatively noninvasive, less operator-dependent, reproducible, provides a quantitative measurement [23,24] and can be performed at the time of CL measurement using the same vaginal probe. Strain elastography is affected by external compression, which means it is highly variable. Conversely, shear wave elastography is significantly affected by target diameter and tissue stiffness. Thus, it may not be applicable for the cervix, which is anisotropic, heterogeneous and a small-sized organ with microstructural complexity [25].

While the prognosis or disease severity is very different depending on gestational age, predicting preterm birth is important in regard to various neonatal morbidities and mortalities. In particular, infants born at <32 weeks of gestation are at a higher risk of not only necrotizing enterocolitis, bronchopulmonary dysplasia or nosocomial infections, but also cerebral palsy (6.3% at 30 weeks and 0.7% at 34 weeks) or mild to severe cognitive impairments (35.3% at 30 weeks and 23.9% at 34 weeks) [26]. Therefore, predicting adverse outcomes before 32 weeks of gestation, by evaluating the efficacy of progesterone in pregnant women through elastography, helps prepare for careful monitoring and intensive neonatal care.

To investigate the possible role of cervical elastography in determining the treatment success of progesterone in pregnant women with short cervix, the present study aimed to assess the efficacy in predicting sPTD at <32 weeks of gestation in singleton pregnancy with an asymptomatic short cervix of ≤2.5 cm at 16–28 weeks of gestation in women who were prescribed progesterone through a comparison of elastographic parameters pre and post progesterone treatment.

## 2. Materials and Methods

### 2.1. Study Population

A Korean prospective observational multicenter cohort study titled “Prediction of PTD and establishing mechanisms of PTD prevention in high-risk pregnancies of PTD using cervical elastography” recruited participants at seven institutions between June 2018 and June 2020 following study approval by the Institutional Review Board, and written informed consent was obtained from subjects prior to enrollment.

From this cohort database, singleton pregnant women diagnosed with a short cervix (≤2.5 cm) at 16 to 28 weeks of gestation, and who consented to progesterone treatment for preterm birth prevention, were extracted for analysis. Women with a CL ≤0.5 cm at the time of enrollment, with multifetal pregnancy, who had received cerclage or progesterone treatment within two weeks before enrollment, or had cerclage placed within one week of progesterone treatment, were excluded.

Progesterone treatment with either daily vaginal micronized natural progesterone (200 mg) at bedtime, or weekly intramuscular injection of 17-hydroxyprogesterone caproate (250 mg), was initiated from the day of diagnosis of a short cervix and continued until 36 weeks of gestation. CL measurement and elastography were performed serially at baseline (preprogesterone, PP0) and at one week postprogesterone treatment (PP1). The participants who delivered before 32 weeks of gestation, due to either spontaneous preterm labor or preterm premature rupture of membranes, were categorized into the delivery ≤ 32 weeks group (study group), while the remaining participants were categorized into the delivery > 32 weeks group. Demographic, sonographic, elastographic and obstetric parameters, as well as delivery outcomes, were retrospectively retrieved from electronic clinical report forms (Medidata Classic Rave^®^, Dassault Systemes, Mason, OH, USA) and used for analysis.

### 2.2. Cervical Length and Elastography Measurement

CL was measured using a vaginal ultrasound (WS80A Ultrasound System, Samsung Medison, Seoul, Korea), which uses a 6-MHz transvaginal probe. After measuring the CL, elastography was performed in the same plane with the same transvaginal probe using the E-cervix^TM^ system (WS80A; Samsung Medison, Seoul, Korea), a quantification tool used to measure the stiffness of the cervix using elastography. Elastography measurements adhered to the standardized protocol previously suggested by our group [27]. In brief, patients were asked to empty their bladder prior to examination. Using the transvaginal probe, CL was measured three times in the mid-sagittal plane of the cervix in which the endocervical canal was clearly delineated with caution not to apply pressure to the cervix. The shortest CL was recorded.

Subsequently, the E-cervix^TM^ program was used for elastography measurements in the same plane. During the acquisition process, the patient was allowed to breathe normally, and the probe was held still until all the motion bars turned green followed by an autofreeze. For the strain measurement, the region of interest (ROI) caliper was placed on the internal and external os and then on each corner of the cervix so that the ROI box included the entire cervix. Cervical strain acquisition was performed three times, and the median value of each strain parameter was used for analysis. Strain parameters were the mean internal os strain (IOS), mean external os strain (EOS), elasticity contrast index (ECI), and hardness ratio (HR). ECI and HR represent the mean strain value within 1.5 cm of the cervical canal, and IOS and EOS represent the mean of strain value within a 1.0-cm radius from the internal and external cervical os, respectively (Figure 1) [8].

### 2.3. Elastography Image Auditing

Two independent investigators (S.Y.O. and H.J.S.) with more than five years of experience using E-cervix^TM^ cross-reviewed all elastography images for appropriateness of image quality and caliper placement. Elastographic measurements were discarded if there was an asymmetrical cervix (anterior:posterior cervix width ratio <50%), a polyp present within the cervical canal, presence of nabothian cyst or nonvisualized internal os due to shadowing [27]. When caliper placement was considered inappropriate, elastography measurement was repeated on elastography data retrieved from the storage server.

### 2.4. Statistical Analysis

Baseline maternal and obstetric parameters and outcomes were compared between the two groups, delivery ≤ 32 weeks and delivery > 32 weeks, using Pearson’s χ^2^ test, Fisher’s exact test, independent two-sample t-test and the Mann-Whitney U test as appropriate. Continuous variables were summarized as median (range) or mean ± standard deviation, and categorical variables as frequency (%). Uni and multivariate logistic regression analyses were used to determine the association between elastographic parameters and sPTD at <32 weeks of gestation. CL was included as a confounding covariate. To evaluate the ability of a parameter, or combinations of parameters, to predict sPTD at <32 weeks of gestation, the areas under the receiver operating characteristic curve (AUC) were calculated and compared using a bootstrap test. Statistical significance was defined as a *p*-value of <0.05. Analyses were performed using R version 3.6.3 (R Foundation for Statistical Computing, Vienna, Austria).

## 3. Results

### 3.1. Study Population and Maternal Characteristics

Of the 1530 participants from the prospective observational multicenter cohort study titled ‘Prediction of preterm delivery and establishing mechanisms of preterm delivery prevention in high-risk pregnancies of preterm delivery using cervical elastography’, a total of 190 pregnant women were diagnosed with short cervix at 16 to 28 weeks of gestation and prescribed progesterone for the prevention of PTD. After excluding 50 twin pregnancies, 12 patients who had cerclage placed, and 13 patients lost to follow-up, 115 were eligible for the final analysis (Figure 2).

The median age of the participants was 34 years (range, 25–42), and the gestational age at enrollment was 22 weeks and 6 days (range, 16 weeks and 1 day–28 weeks and 6 days) on average. The median CL measured at enrollment was 22 mm (10–25 mm) and funneling was present in 7.0% (8/115) of cases. The incidence of sPTD at <32 weeks of gestation was 7.8% (9/115).

No significant differences in maternal demographics and clinical characteristics were observed between the two groups of delivery at ≥32 weeks and delivery at <32 weeks. The gestational age, CL and presence of funneling at the time of enrollment did not differ between groups. In terms of delivery outcomes, the group of delivery <32 weeks had lower birth weight and higher rate of neonatal intensive care unit admission than the group of delivery ≥32 weeks (Table 1).

### 3.2. Pre and Postprogesterone Elastographic Parameters

Overall, no significant change was found in the cervical elastographic parameters measured at baseline and at one week after the progesterone treatment (Appendix A). However, there was a trend for IOS, EOS, and ECI to be increased and HR to be decreased in the group of delivery <32 weeks compared to that in the group of delivery ≥32 weeks, but this did not reach statistical significance (Figure 3).

### 3.3. Association between Elastographic Parameters and sPTD at <32 Weeks of Gestation

Table 2 presents the pre and postprogesterone treatment elastographic parameters. When compared to the group of delivery ≥32 weeks, the group of delivery <32 weeks had higher IOS (*p* = 0.001) and ECI (*p* = 0.037), lower HR (*p* = 0.02) at baseline, and a significantly higher IOS, EOS and ECI (*p* = 0.002, 0.018, and 0.02) and lower HR (*p* = 0.004) at one week postprogesterone treatment.

A CL-adjusted logistic regression revealed that IOS at baseline and IOS, EOS and HR at one week postprogesterone treatment were significantly associated with sPTD at <32 weeks of gestation (Table 3). Furthermore, the odds of sPTD at <32 weeks of gestation increased by 2.92, 4.39 and 3.65-fold, respectively, for each 0.1 increase in IOS at baseline and IOS and EOS at one week postprogesterone treatment (adjusted for CL; *p* = 0.04, 0.012 and 0.026, respectively).

### 3.4. Prediction of sPTD at <32 Weeks of Gestation Based on Clinical and Elastographic Parameters

As shown in Table 4, CL at diagnosis and elastographic parameters associated with sPTD at <32 weeks were used to evaluate the predictive ability of the models. HR was not included in the combination as the association was weak (OR = 0.92, *p* = 0.016). IOS and EOS from the same time point were not combined as the strain area of IOS and EOS overlap in cases of short cervix.

As a result, the combination of CL at diagnosis, IOS at baseline (PP0 IOS) and EOS at one week postprogesterone treatment (PP1 EOS) showed a significantly higher AUC than CL alone (0.858, *p* = 0.041; Figure 4).

Furthermore, the following formula was developed to predict the probability of sPTD at <32 weeks of gestation in pregnant women with a short cervix treated with progesterone based on CL at diagnosis, PP0 IOS and PP1 EOS:(1)$$P\left(y=1\right)=\frac{\exp\left(-2.23+0.82\times \mathrm{PP}0\;\mathrm{IOS}/10+1.02\times \mathrm{PP}1\;\mathrm{EOS}/10-0.29\times \mathrm{CL}\right)}{1+\exp\left(-2.23+0.82\times \mathrm{PP}0\;\mathrm{IOS}/10+1.02\times \mathrm{PP}1\;\mathrm{EOS}/10-0.29\times \mathrm{CL}\right)}$$

Using 0.032 as a threshold, the sensitivity and specificity for predicting sPTD at <32 weeks were 80.0% and 70.0%, respectively, with a PPV of 12.9% and NPV of 97.7%.

## 4. Discussion

To the best of our knowledge, this is the first study to demonstrate the efficacy of cervical elastographic parameters in detecting changes after progesterone treatment for preventing sPTD in singleton pregnancies with a short cervix. The major findings of our study are as follows: in singleton pregnancies with a short cervix on progesterone treatment for sPTD prevention: (1) higher IOS, ECI and lower HR at pretreatment (PP0), and higher IOS, EOS, ECI and lower HR at one week post-treatment (PP1) were associated with sPTD at <32 weeks; (2) higher IOS at PP0 and higher IOS and EOS at PP1 were associated with a 2.92, 4.39 and 3.65-fold increase in the risk of sPTD at <32 weeks, respectively; (3) a combination of cervical elastographic parameters (IOS at PP0 and EOS at PP1) with CL at diagnosis demonstrated a significantly higher predictive ability than CL alone for predicting sPTD at <32 weeks.

Progesterone prevents cervical softening by modifying cytokines, regulating the presence of macrophages in the stroma and altering the extracellular matrix in the cervix [20,21]. Therefore, it is widely used in clinical practice to prevent PTD in pregnant women with prior PTD or a short cervix at the mid-trimester [2,19,22,28,29,30]. However, predicting PTD by differentiating women who have a good response to progesterone from those who do not respond well, eventually leading to PTD, is difficult. Additional ultrasound parameters such as sludge, fetal fibronectin or inflammatory markers and cytokines, are considered candidates for adjuvant modalities, but their efficacy is not sufficient for clinical use [31,32].

Studies on elastography for predicting PTD in women with a short cervix are limited. Wozniak et al. showed, in a study of 109 women with CL ≤25 mm, that elastographic assessment of the internal os by color mapping at 18–22 weeks of gestation was significantly associated with the risk of PTD [33]. Hernandez-Andrade et al. demonstrated the feasibility of cervical elastography in predicting sPTD in 28 singleton pregnancies with short cervix diagnosed at 18–24 weeks of gestation and treated with daily vaginal progesterone [34]. These studies are in accordance with our finding that elastography can help identify pregnant women with a short cervix at a high risk for PTD [7,8,33,34]. However, our study is distinct from previous studies in that we evaluated the strain in the entire cervix, not only in the internal os, we assessed changes in strain due to physiological internal pulsation rather than with an externally applied strain such as a shear wave, we serially evaluated the cervical strain and we showed changes before and after progesterone treatment rather than at a single time point.

Park et al. conducted elastography measurements in 130 pregnant women with short cervix using the same technique as in our study [8]. However, they did not find a significant association between IOS, EOS and HR and sPTD. In contrast, we found that higher IOS and EOS, and lower HR, were associated with an increased risk of sPTD. This discrepancy in the results may be first attributed to the distinct study population and to the different end-points used in these studies. Park et al. included pregnant women with short CL regardless of progesterone use, and their primary outcome was sPTD at <37 weeks, whereas all patients in our study received progesterone treatment focusing on early preterm birth at <32 weeks of gestation. Second, the E-cervix^TM^ program applied in our study had an updated form compared to the one used in the previous study. The present version was modified to enhance sensitivity of the motion bar that monitors for the patient’s and fetus’ motions to reduce measurement errors driven by external factors. Third, a standardized protocol for measuring cervical elastography using the E-cervix^TM^ program was formulated and implemented across participating institutions during the study period.

Among the prediction models for sPTD at <32 weeks of gestation, the model that employed a combination of CL and IOS at diagnosis, and EOS at one week postprogesterone treatment, showed the highest efficiency. Strain of the internal cervical os has been consistently found to be associated with sPTD in a singleton pregnancy with short cervix [33,34]. Interestingly, we found that the strain of the external cervical os one week after progesterone treatment was also highly associated. As progesterone was administered vaginally in 95.7% of the patients, it can be speculated that the progesterone suppository inserted into the vagina makes direct contact with the external os, where the progesterone-driven cervical matrix modification should be most active, explaining why EOS was significant in determining progesterone-responsive cervices. Thus, pregnant women who do not show improvement in IOS and EOS after one week may undergo progressive cervical softening.

The strength of this study is that it is the first to evaluate changes in cervical stiffness after using progesterone in a longitudinal manner in pregnant women with short CL. Moreover, to maintain objectivity and obtain reliable elastographic measurements, a standardized protocol was adhered to and image auditing was applied prior to data analysis. However, we did not perform a formal sample size calculation in this study. At the beginning of the study, various clinical factors such as the accuracy of the cervical elastography and the dropout rate, were not clearly established. Therefore, it was difficult to determine an adequate sample size. Additionally, our study is limited in that the number of cases included in the study group was small, which explains the low PPV of the prediction model. As demonstrated in the PREGNANT trial, in which the incidence of sPTD at <33 weeks was only 8.9% (21/235) [35], early PTD in singleton pregnancies receiving progesterone due to short cervix is uncommon. There is a probability that a type 1 error, or chance, was the reason for the reported association between the cervical elastography parameters and PTD at <32 weeks given our small sample size and missing data. Therefore, studies including a larger number of cases may be necessary to further substantiate our findings.

## 5. Conclusions

In this study, the combination of CL and IOS at baseline and EOS at one week postprogesterone treatment predicts sPTD at <32 weeks better than CL alone in singleton pregnancies with short cervix in women receiving progesterone. Therefore, performing cervical elastography before and after progesterone treatment may be useful in identifying patients at high risk for sPTD at <32 weeks of gestation.

## Figures and Tables

**Figure 1 ijerph-18-02026-f001:**
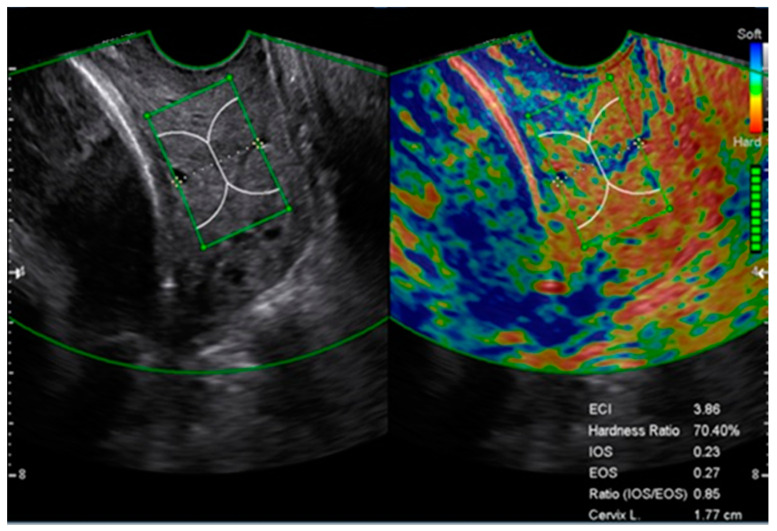
Cervical elastography image obtained from a short cervix using E-cervix^TM^. Acquisition of cervical elastography was performed in the mid-sagittal plane when measuring cervical length. Elastographic parameters and cervical length (CL) are displayed on the bottom left of the image. ECI, elasticity contrast index; EOS, mean of external os strain; HR, hardness ratio; IOS, mean of internal os strain.

**Figure 2 ijerph-18-02026-f002:**
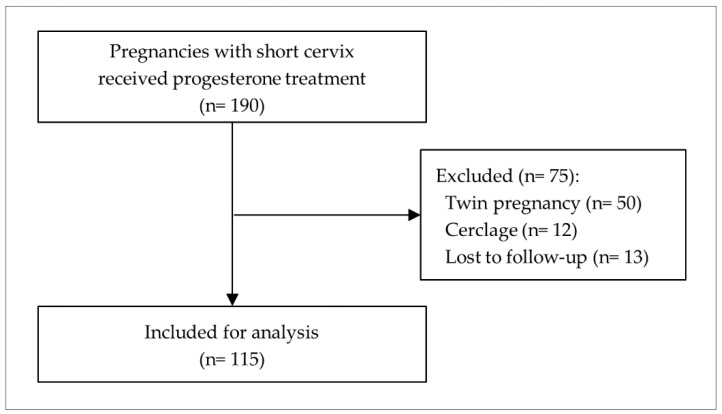
Study population.

**Figure 3 ijerph-18-02026-f003:**
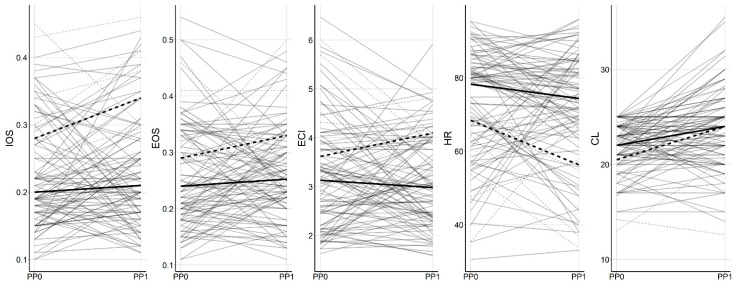
Preprogesterone and postprogesterone changes in elastographic parameters. Solid and dashed lines represent the mean value of the groups of delivery ≥32 weeks and delivery <32 weeks, respectively. IOS, mean of internal os strain; EOS, mean of external os strain; ECI, elasticity contrast index; HR, hardness ratio; CL, cervical length; PP0, preprogesterone; PP1, 1 week postprogesterone.

**Figure 4 ijerph-18-02026-f004:**
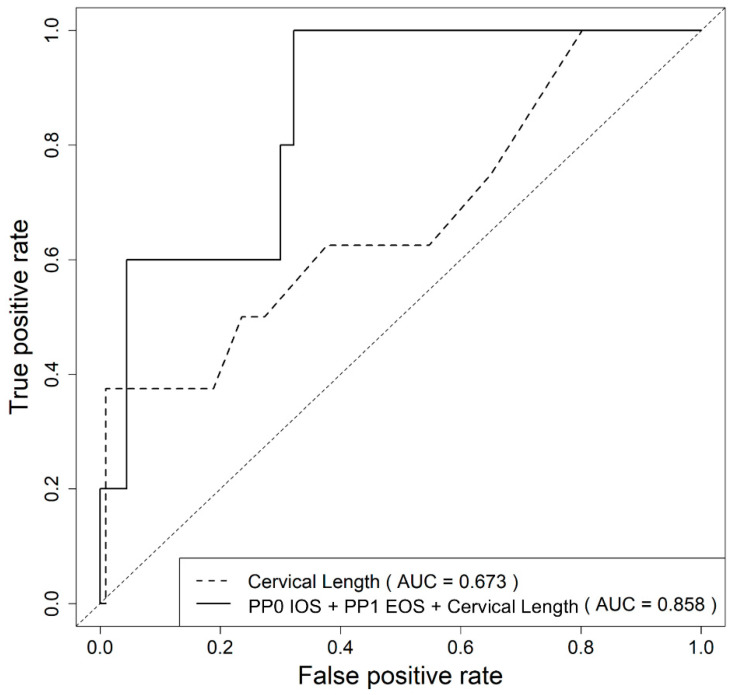
Receiver operating characteristic curve for the prediction of spontaneous preterm delivery at <32 weeks using the predicted probability calculated from the logistic regression model using cervical length and cervical elastographic parameters. IOS, mean internal os strain; EOS, mean external os strain; PP0, preprogesterone; PP1, 1 week postprogesterone treatment.

**Table 1 ijerph-18-02026-t001:** Baseline characteristics, clinical findings, and pregnancy outcomes of the study population.

Variables	Total (*n* = 115)	Delivery ≥32 Weeks (*n* = 106)	Delivery <32 Weeks (*n* = 9)	*p*-Value
Maternal age, years	34 (25–42)	34 (25–42)	35 (28–39)	0.415
Prepregnancy BMI, kg/m^2^	22.9 (19.0–36.2)	22.9 (19.0–36.2)	22.8 (20.4–27.3)	0.577
Current smoking	1 (0.9%)	1 (0.9%)	0 (0.0%)	1.000
Multiparity	57 (49.6%)	54 (50.9%)	3 (33.3%)	0.655
Prior miscarriage	39 (33.9%)	35 (33.0%)	4 (44.4%)	0.803
Prior cesarean section	20 (17.4%)	19 (17.9%)	1 (11.1%)	0.999
Prior preterm birth	8 (7.0%)	8 (7.5%)	0 (0.0%)	0.999
IVF-ET	17 (14.8%)	14 (13.2%)	3 (33.3%)	0.227
Prior conization	10 (8.7%)	9 (8.5%)	1 (11.1%)	0.844
Pregestational diabetes mellitus	2 (1.7%)	1 (0.9%)	1 (11.1%)	0.192
GDM	11 (9.9%)	10 (9.7%)	1 (12.5%)	0.852
Preeclampsia	2 (1.8%)	1 (1.0%)	1 (12.5%)	0.182
Chronic hypertension	3 (2.6%)	2 (1.9%)	1 (11.1%)	0.299
Thyroid disease	6 (5.2%)	4 (3.8%)	2 (22.2%)	0.098
GA at diagnosis	22 w 6 d (16 w 1 d–28 w 6 d)	22 w 5 d (16 w 1 d–28 w 6 d)	23 w 5 d (17 w 0 d–26 w 3 d)	0.599
Cervical length, mm	22.0 (10.0–25.0)	22.0 (10.0–25.0)	20.5 (13.0–24.0)	0.103
Funneling	8 (7.0%)	6 (5.7%)	2 (22.2%)	0.176
Positive fibronectin	27 (35.5%)	22 (32.4%)	5 (62.5%)	0.256
Progesterone type				
Vaginal suppository	110 (95.7%)	101 (95.3%)	9 (100.0%)	0.999
IM injection	5 (4.3%)	5 (4.7%)	0 (0.0%)	
GA at delivery	38 w 2 d (18 w 5 d–41 w 0 d)	38 w 3 d (33 w 1 d–41 w 0 d)	27 w 0 d (18 w 5 d–31 w 6 d)	<0.001
Cesarean delivery	62 (53.9%)	55 (51.9%)	7 (77.8%)	0.337
Birth weight, g	3040 (830–4030)	3110 (1760–4030)	1075 (830–1830)	<0.001
Apgar score <7 at 5 min	3 (2.9%)	2 (2.1%)	1 (12.5%)	0.296
NICU admission	21 (20.4%)	13 (13.7%)	8 (100.0%)	<0.001

Data are presented as the median (range) or number (%). BMI, body mass index; IVF-ET, in vitro fertilization-embryo transfer; GDM, gestational diabetes; GA, gestational age; IM, intramuscular; NICU, neonatal intensive care unit; w, weeks; d, days.

**Table 2 ijerph-18-02026-t002:** Comparison of pre and postprogesterone elastographic parameters between the groups of delivery ≥32 weeks and delivery <32 weeks.

Variables	PP0 (Pretreatment)		*p*-Value	PP1 (Post-Treatment)		*p*-Value
	Delivery ≥ 32 Weeks (*n* = 105)	Delivery < 32 Weeks (*n* = 9)		Delivery ≥ 32 Weeks (*n* = 90)	Delivery < 32 Weeks (*n* = 6)	
IOS	0.22 ± 0.07	0.30 ± 0.09	0.001	0.23 ± 0.08	0.34 ± 0.09	0.002
EOS	0.26 ± 0.09	0.32 ± 0.08	0.063	0.26 ± 0.08	0.35 ± 0.09	0.018
ECI	3.19 ± 1.09	4.00 ± 1.23	0.037	3.09 ± 0.95	4.03 ± 0.58	0.020
HR	73.19 ± 14.77	61.17 ± 12.74	0.020	72.29 ± 15.40	53.24 ± 14.98	0.004

Data are presented as mean ± SD. PP0, baseline; PP1, post-treatment 1 week; IOS, mean of internal os strain; EOS, mean of external os strain; ECI, elasticity contrast index; HR, hardness ratio.

**Table 3 ijerph-18-02026-t003:** Association between elastographic parameters and spontaneous preterm delivery at < 32 weeks of gestation.

Variable	PP0 (Pretreatment)				PP1 (Post-Treatment)			
	Unadjusted OR (95% CI)	*p*-Value	Adjusted OR (95% CI) *	*p*-Value	Unadjusted OR (95% CI)	*p*-Value	Adjusted OR (95% CI) *	*p*-Value
IOS (×0.1)	3.60 (1.56–9.28)	0.004	2.92 (1.07–8.77)	0.040	3.94 (1.55–12.46)	0.007	4.39 (1.52–17.29)	0.012
EOS (×0.1)	1.89 (0.94–3.87)	0.071	2.13 (0.94–5.03)	0.070	2.88 (1.16–7.97)	0.027	3.65 (1.28–13.74)	0.026
ECI (×0.1)	1.06 (1.00–1.12)	0.045	1.06 (0.99–1.13)	0.089	1.11 (1.01–1.23)	0.032	1.11 (1.00–1.25)	0.054
HR	0.95 (0.91–0.99)	0.027	0.96 (0.91–1.01)	0.122	0.93 (0.88–0.98)	0.012	0.92 (0.86–0.98)	0.016

PP0, baseline; PP1, 1 week postprogesterone; OR, odds ratio; CI, confidence interval; IOS, mean of internal os strain; EOS, mean of external os strain; ECI, elasticity contrast index; HR, hardness ratio. * Adjusted for cervical length at diagnosis.

**Table 4 ijerph-18-02026-t004:** Comparison of area under the ROC curve between CL at diagnosis alone and combination models for the prediction of spontaneous preterm delivery at <32 weeks of gestation in women with a short cervix treated with progesterone.

Model	AUC (95% CI)	*p*-Value *
CL	0.673 (0.448–0.898)	Reference
PP0 IOS (×0.1) + CL	0.794 (0.607–0.981)	0.105
PP1 IOS (×0.1) + CL	0.831 (0.575–1.000)	0.133
PP1 EOS (×0.1) + CL	0.827 (0.624–1.000)	0.103
PP0 IOS (×0.1) + PP1 IOS (×0.1) + CL	0.831 (0.561–1.000)	0.149
PP0 IOS (×0.1) + PP1 EOS (×0.1) + CL	0.858 (0.714–1.000)	0.041

ROC, receiver operating characteristic curve; CI, confidence interval; CL, cervix length at diagnosis; IOS, mean of internal os strain; EOS, mean of external os strain; PP0, baseline; PP1, 1 week postprogesterone. * Comparison between CL and other models.

## Data Availability

The datasets generated during and analyzed in the current study cannot be publicly available however, can be made available upon the request made to the corresponding author on reasonable request following the IRB approval.

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
