# Peer review of "The Feasibility of Cervical Elastography in Predicting Preterm Delivery in Singleton Pregnancy with Short Cervix Following Progesterone Treatment"

_ijerph, 2021, doi:10.3390/ijerph18042026_

Round 1

Reviewer 1 Report

Review of Feasibility of cervical elastography in predicting preterm delivery in singleton pregnancy with short cervix after progesterone treatment.

The objective of this study was to determine whether cervical elastography done before and after treatment with progesterone predicts preterm delivery before 32 weeks gestation in women with a short cervical length. A secondary objective was to assess the usefulness of cervical elastography parameters in predicting PTD < 32 weeks of gestation. The second objective appears to be unnecessary since the pre-treatment data is being used to answer this objective; and the primary objective is to evaluate the usefulness of the pretreatment and post-treatment data to predict preterm deliveries < 32 weeks.  In other words the secondary objective is included in the primary objective. I suggest limiting the manuscript to one objective. They reported that a regression analysis adjusted for cervical length at diagnosis showed an association between cervical elastography parameters and spontaneous preterm delivery before 32 weeks gestation.

The words “despite treatment” in the last sentence of the abstract (line 42) need to be deleted since elastography parameters pre-treatment were also reported to be different. The first paragraph of their introduction must be re-written since it has misleading statements. The statement that two-thirds of the short cervix population do not benefit from progesterone treatment is false (line 51-52). The referenced article is referring to pregnancies with a history of prior preterm deliveries. It is not addressing the efficacy of progesterone on cervical shortening. It states “at best progesterone supplementation prevents only one-third of recurrent preterm births”.  Please note the words recurrent preterm births. The sentence stating “conflicting reports regarding the efficacy of progesterone in pregnant women with a short cervix have been reported” is misleading and may be written better since only one of the reference studies used cervical shortening as the sole inclusion criteria (line 52-54).  The OPPTIMUM study reported that vaginal progesterone was not associated with a reduction of preterm births. The inclusion criteria was history of preterm birth, second trimester loss, premature rupture of membranes, history of cervical procedures, and positive fetal fibronectin test. The criteria was revised after approximately 1 1/2 years of enrollment to include women with a cervical length ≤ 25 mm. The subgroup analysis of patients with cervical shortening did not show benefit. The Hassan study only used a short cervical length as the inclusion criteria and reported a reduction in preterm birth with vaginal progesterone treatment.  

Why did the authors select PTD < 32 weeks gestation as their primary objective? Why not < 34 or < 37 weeks? Would results differ if 34 or 37 weeks are used? Please discuss. The authors state that progesterone treatment was initiated from day of diagnosis. In my practice and the practice of all my colleagues it takes a few days from diagnosis and prescription of medication to get the medication and start treatment. How was it possible to start progesterone treatment on the day of diagnosis? Hard to believe!

Please change word “experts” to “investigators” (line 153). The numbers on line 183; should read 34 (range 25 - 42). Please clarify (16+1 weeks- 28+6 weeks) on line 185, not sure what it represents; possibly (range 16 weeks and 1 day – 28 weeks and 6 days).

I recommend deleting the statement that “To our knowledge, this is the first study” and rewording the sentence (line264). This statement is presumptuous on their part. In addition, they reference studies where cervical elastography was done on patients with short cervices that were treated with progesterone.

Tables need to be properly label. There should be no control group or study group based on their study design. They state that this is a prospective observational cohort study.  Therefore, the comparisons in the Tables should be between Deliveries > 32 weeks and Deliveries ≤ 32 weeks (primary outcome of study). The authors have incorrectly label the comparison groups as study group and control group and must revise the Tables accordingly. The words control and study groups must also be removed from other sections of the manuscript. In Table 1 the GA at delivery, weeks needs revision. They are reporting the median (range) not sure what 38 +2 means and what (18 +5 – 41 +0) means. Does it mean 38 weeks 2 days (18 weeks 5 days - 41 weeks 0 days)? If so, maybe it can be written as (18w 5d – 41w 0d). Table 2 does not state how the data is presented. Is it Mean ±SD? Is it median? Please add data presentation format to Table 2.

The word “was” on line 269 should be changed to “were”. The sentence “The strength of the study is that this is the first study to demonstrate…” (line 332-334) needs to be remove since the findings of a study are never a strength of a study, they are only findings. A major weakness of the study is the absence of a sample size calculation prior to data analysis and this fact must be included in the discussion.  All cohort studies with primary objectives should have sample size calculations before data analysis.  The authors correctly note that a weakness of their study is the small sample size of patients with delivery < 32 weeks gestation. They should report that the small sample size can cause the results to have a type 1 error.  Another weakness to state in their discussion is that the cervical elastography parameters 1 week after progesterone treatment (PP1) were not obtained in all of the participants and in the group with a PTD < 32 weeks 3 out the 9 women had PP1 data missing. They should have a sentence stating that “There is a probability that a type 1 error or chance is the reason for the reported association between the cervical elastography parameters and PTD < 32 weeks because of our small sample size and missing data. “

The authors cannot state that they are reporting progesterone driven changes of cervical stiffness. How do they know what caused the changes? They should remove this statement (line 332). The changes could be the result of the natural progression of the underlying disease process, or other intrinsic or extrinsic factors. The first sentence in the conclusion section is poorly written and confusing. Please re-write conclusion section. The words “despite treatment” should be removed from the last sentence (line 347).

Author Response

Response to Reviewer 1 Comments

Point 1: The objective of this study was to determine whether cervical elastography done before and after treatment with progesterone predicts preterm delivery before 32 weeks gestation in women with a short cervical length. A secondary objective was to assess the usefulness of cervical elastography parameters in predicting PTD < 32 weeks of gestation. The second objective appears to be unnecessary since the pre-treatment data is being used to answer this objective; and the primary objective is to evaluate the usefulness of the pretreatment and post-treatment data to predict preterm deliveries < 32 weeks.  In other words the secondary objective is included in the primary objective. I suggest limiting the manuscript to one objective. They reported that a regression analysis adjusted for cervical length at diagnosis showed an association between cervical elastography parameters and spontaneous preterm delivery before 32 weeks gestation.

Response 1:  Thank you for your recommendation. In order to clarify the aim of the study, we revised the text as follows:

“To investigate the possible role of cervical elastography in determining the treatment success of progesterone in pregnant women with short cervix, the present study aimed to assess the efficacy in predicting sPTD at <32 weeks of gestation in singleton pregnancy with an asymptomatic short cervix of ≤2.5 cm at 16-28 weeks of gestation in women who were prescribed progesterone through a comparison of elastographic parameters pre- and post-progesterone treatment.”

Point 2: The words “despite treatment” in the last sentence of the abstract (line 42) need to be deleted since elastography parameters pre-treatment were also reported to be different. The first paragraph of their introduction must be re-written since it has misleading statements. The statement that two-thirds of the short cervix population do not benefit from progesterone treatment is false (line 51-52). The referenced article is referring to pregnancies with a history of prior preterm deliveries. It is not addressing the efficacy of progesterone on cervical shortening. It states “at best progesterone supplementation prevents only one-third of recurrent preterm births”. Please note the words recurrent preterm births. The sentence stating “conflicting reports regarding the efficacy of progesterone in pregnant women with a short cervix have been reported” is misleading and may be written better since only one of the reference studies used cervical shortening as the sole inclusion criteria (line 52-54). The OPPTIMUM study reported that vaginal progesterone was not associated with a reduction of preterm births. The inclusion criteria was history of preterm birth, second trimester loss, premature rupture of membranes, history of cervical procedures, and positive fetal fibronectin test. The criteria was revised after approximately 1 1/2 years of enrollment to include women with a cervical length ≤ 25 mm. The subgroup analysis of patients with cervical shortening did not show benefit. The Hassan study only used a short cervical length as the inclusion criteria and reported a reduction in preterm birth with vaginal progesterone treatment. 

Response 2: Thank you for your comments. First, we deleted the words “despite treatment” as per your recommendation. Second, regarding the first paragraph of the introduction, the study of Norwitz et al. provides an explanation about progesterone supplementation in women with cervical shortening, while Fonseca et al. found 24% of preterm births before 34 weeks of gestation in the progesterone group, and Hassan et al. reported that the percentage of preterm births in the progesterone group ranged from 5.1% to 14.5%. We focused on the fact that preterm birth still occurs despite progesterone treatment in women with a short cervix. In order to improve clarity, we changed the introduction as follows:

“Nonetheless, preterm birth occurs in one-third of women with short cervix undergoing progesterone treatment [3]. Conflicting results regarding the efficacy of progesterone in pregnant women with a short cervix have been reported depending on the involved risk factors [1,4]. Plausible reasons may be as follows; 1) cervical length (CL) alone does not yield high positive predictive value (PPV) in predicting sPTD [5,6]; 2) the mechanism of cervical shortening maybe different from the target mechanism of progesterone; 3) patients who may benefit from progesterone were not carefully selected as valid test to screening is lacking. Therefore, a clinical test that can predict whether progesterone treatment will be successful in patients with short cervix would help more selectively differentiate those who would benefit with progesterone treatment from the non-responsive group, for which other treatment options, including cerclage, should be considered for preventing sPTD.”

Point 3: Why did the authors select PTD < 32 weeks gestation as their primary objective? Why not < 34 or < 37 weeks? Would results differ if 34 or 37 weeks are used? Please discuss.

Response 3: We have described the rationale behind choosing PTD <32 weeks as the primary objective in the discussion of this article. Analyses concerning the prediction of PTD at <34 and <37 weeks were also conducted. There were significant results in these gestational age groups. However, compared to late preterm birth over 34 weeks, studies reported that preterm birth at <32 weeks had a greater impact on the incidence of complications related with prematurity. Since the incidence of long-term complications increases in infants who delivered before 32 weeks of gestation, early prediction and efforts to prevent premature birth are important for improving neonatal outcomes in consideration of social and medical costs. Therefore, we used 32 weeks as the PTD threshold.

Point 4: The authors state that progesterone treatment was initiated from day of diagnosis. In my practice and the practice of all my colleagues it takes a few days from diagnosis and prescription of medication to get the medication and start treatment. How was it possible to start progesterone treatment on the day of diagnosis? Hard to believe!

Response 4: The National Health Insurance program in South Korea has improved accessibility and equity in health care utilization and improved the level of health for the insured person. In Korea, it is possible to see a doctor and purchase medications prescribed by the doctor on the same day. Therefore, for high-risk pregnant women whose cervix length was less than 2.5 cm without uterine contractions, progesterone treatment can be started within 24 hours after visiting the outpatient clinic (the day of diagnosis).

Point 5: Please change word “experts” to “investigators” (line 153). The numbers on line 183; should read 34 (range 25 - 42). Please clarify (16+1 weeks- 28+6 weeks) on line 185, not sure what it represents; possibly (range 16 weeks and 1 day – 28 weeks and 6 days).

Response 5:  As the reviewer mentioned, we changed the results according to the reviewer’s suggestion as follows :

(line 162) Two independent investigators (S.Y.O. and H.J.S) ...

(line 194-196) The median age of participants was 34 years (range 25-42), and the gestational age at enrollment was 22 weeks and 6 days (range 16 weeks and 1 day–28 weeks and 6 days) on average.

Point 6: I recommend deleting the statement that “To our knowledge, this is the first study” and rewording the sentence (line 264). This statement is presumptuous on their part. In addition, they reference studies where cervical elastography was done on patients with short cervices that were treated with progesterone.

Response 6: As the reviewer mentioned, we agree that there have been other studies that have performed elastography in a group using progesterone with a short cervical length. Most studies using cervical elastography reported the efficacy for predicting preterm birth, but there are no studies comparing the elastographic parameters pre- and post- progesterone treatment serially and evaluating the efficacy of elastography for predicting treatment success. Other studies have also measured cervical elastography only at one time point. Conversely, we conducted a prospective study in which we performed cervical elastography before and 1 week after progesterone use in pregnant women with short cervix. The differences in elastrographic parameters before and after progesterone treatment were investigated in the same patient in a repeated measures design. To improve clearly, we changed the sentence as follows :

 “To the best of our knowledge, this is the first study to demonstrate the efficacy of cervical elastographic parameters in detecting changes after progesterone treatment for preventing sPTD in singleton pregnancies with a short cervix.”

Point 7: Tables need to be properly label. There should be no control group or study group based on their study design. They state that this is a prospective observational cohort study.  Therefore, the comparisons in the Tables should be between Deliveries > 32 weeks and Deliveries ≤ 32 weeks (primary outcome of study). The authors have incorrectly label the comparison groups as study group and control group and must revise the Tables accordingly. The words control and study groups must also be removed from other sections of the manuscript. In Table 1 the GA at delivery, weeks needs revision. They are reporting the median (range) not sure what 38 +2 means and what (18 +5 – 41 +0) means. Does it mean 38 weeks 2 days (18 weeks 5 days - 41 weeks 0 days)? If so, maybe it can be written as (18w 5d – 41w 0d). Table 2 does not state how the data is presented. Is it Mean ±SD? Is it median? Please add data presentation format to Table 2.

Response 7: As per the reviewer’s recommendation, we changed ‘control group’ to ‘delivery ≥32 weeks’. and ‘sPTD (study) group’ to ‘delivery <32 weeks’. In addition, we restated the gestational age data in the table and added data presentation under Table 2 as follows:

Data are presented as mean ± SD

Point 8: The word “was” on line 269 should be changed to “were”. The sentence “The strength of the study is that this is the first study to demonstrate…” (line 332-334) needs to be remove since the findings of a study are never a strength of a study, they are only findings. A major weakness of the study is the absence of a sample size calculation prior to data analysis and this fact must be included in the discussion. All cohort studies with primary objectives should have sample size calculations before data analysis. The authors correctly note that a weakness of their study is the small sample size of patients with delivery < 32 weeks gestation. They should report that the small sample size can cause the results to have a type 1 error.  Another weakness to state in their discussion is that the cervical elastography parameters 1 week after progesterone treatment (PP1) were not obtained in all of the participants and in the group with a PTD < 32 weeks 3 out the 9 women had PP1 data missing. They should have a sentence stating that “There is a probability that a type 1 error or chance is the reason for the reported association between the cervical elastography parameters and PTD < 32 weeks because of our small sample size and missing data. “

Response 8:

  1. As the reviewer mentioned, we change the word “was” to “were” on line 281.
  2. This is an observational study; since few studies have included only women who used progesterone for short cervix, it was not feasible to perform a formal statistical sample size calculation for this study. We added information about the absence of a priori sample size calculation and possibility of type 1 error in the discussion as follows :

 “However, we did not perform a formal sample size calculation in this study. At the beginning of the study, various clinical factors such as the accuracy of the cervical elastography and the dropout rate were not clearly established; therefore, it was difficult to determine the adequate sample size.”

“There is a probability that a type 1 error or chance was the reason for the reported association between the cervical elastography parameters and PTD at < 32 weeks given our small sample size and missing data.”

Point 9: The authors cannot state that they are reporting progesterone driven changes of cervical stiffness. How do they know what caused the changes? They should remove this statement (line 332). The changes could be the result of the natural progression of the underlying disease process, or other intrinsic or extrinsic factors. The first sentence in the conclusion section is poorly written and confusing. Please re-write conclusion section. The words “despite treatment” should be removed from the last sentence (line 347).

Response 9:

  1. Thank you for the recommendations. We agree that we cannot definitively attribute the changes in cervical stiffness to the effect of progesterone. However, we still highlight that this is the first paper to visually analyze changes in elasticity by measuring cervical elastography before and after progesterone use. Thus, we revised the sentence in the discussion as follows :

“The strength of this study is that this is the first to evaluate changes in cervical stiffness after using progesterone in a longitudinal manner in pregnant women with short CL.”

  1. We revise the first sentence in the conclusion as follows :

 “In this study, the combination of CL and IOS at baseline and EOS at 1-week post progesterone treatment predicts sPTD at <32 weeks better than CL alone in singleton pregnancies with short cervix in women receiving progesterone.”

  1. As the reviewer mentioned, we deleted the words “despite treatment” on line 356.

Reviewer 2 Report

The study aims to evaluate pre- and post-progesterone cervical elastographic parameters in singleton pregnancy receiving progesterone for short cervix of ≤ 2.5 cm between 16 and 28 weeks’ gestation and to assess the efficacy of elastography in predicting spontaneous preterm delivery (sPTD) <32 weeks’ gestation. Among 115 participants eligible for analysis, 9 had sPTD <32 weeks. The authors found that pre-progesterone (PP0) mean of internal os strain (IOS), elasticity contrast index (ECI), and hardness ratio (HR) and 1 week post-progesterone (PP1) IOS, mean of external os strain (EOS), ECI and HR were significantly different between groups. Higher PP0 IOS, PP1 IOS and PP1 EOS were associated with a 2.92-, 4.39- and 3.65-fold increase in the risk of sPTD <32 weeks, respectively (adjusted for CL at diagnosis, all p<0.05, respectively). A combination of CL at diagnosis, PP0 IOS, and PP1 EOS showed significantly higher area under receiver operating characteristic curve (AUC) of 0.858 than that of CL alone (p=0.041). In general, the manuscript is well-written. However, there were some minor points that could be clarified for a better understanding of the readers.

  1. Cervical elastrography is a novel concept for most readers. For clarification, the authors may consider to add some descriptions and to cite more references in the section of Introduction, in addition to the current paragraph in Line 82-88.
  2. The manuscript may be reviewed by a native English speaker.
  3. Some minor points: for example,                                                                               “The” present study (Line 30)

                              “A” combination (Line 38)

                              “were” 80%... (Line 261)

Author Response

Response to Reviewer 2 Comments

Point 1: Cervical elastrography is a novel concept for most readers. For clarification, the authors may consider to add some descriptions and to cite more references in the section of Introduction, in addition to the current paragraph in Line 82-88.

Response 1: Thank you for your comments. We added an explanation about cervical elastography in the introduction as follows :

“Cervical elastography assesses changes in consistency during the course of the maturation process prior to delivery, such as ripening, effacement, and dilatation. Cervical elastography provides information about cervical stiffness and helps predict preterm delivery or successful labor induction [17].”

“Most studies on cervical elastography have evaluated the association between short cervix, regardless of receiving progesterone, and preterm birth [9,10,18]. These studies demonstrated that in pregnant women with short CL, the PPV for predicting preterm birth can be enhanced by adding cervical stiffness rather than using CL alone. However, its feasibility for predicting sPTD in the population with short cervix receiving progesterone treatment is unclear. The theoretical mechanism of progesterone is hypothesized to be through the inhibition of cervical ripening by controlling cervical ECM degradation and remodeling via anti-inflammatory activity and suppression of matrix metalloproteinase inhibitors [19-22]. Thus, progesterone-driven cervical changes may be detected by measuring cervical stiffness in women with short cervix who respond to progesterone treatment..”

Point 2: The manuscript may be reviewed by a native English speaker.

Response 2: As the reviewers suggested, we submitted the manuscript for further proofreading. Thank you for your patience.

Point 3: Some minor points: for example,                                                                               “The” present study (Line 30)   “A” combination (Line 38)   “were” 80%... (Line 261)

Response 3: As the reviewer mentioned, we added these articles and changed the words.

Reviewer 3 Report

The manuscript is a prospective study with the primary objective to evaluate pre- and post-progesterone cervical elastographic parameters in singleton pregnancy receiving progesterone for short cervix of ≤ 2.5 cm between 16 and 28 weeks’ gestation, the second objective was to assess the efficacy of elastography in predicting sPTD <32 weeks’ gestation. The authors reported that a regression analysis adjusted for cervical length at diagnosis showed an association between cervical elastography parameters and spontaneous preterm delivery before 32 weeks gestation.

The article is well written and the topic falls within the scope of the journal and it is of high interest. I would suggest a minor revision before accepting the paper: 

  • 1.      Please, shorten the discussion
  • 2.     Why did the authors select PTD < 32 weeks gestation as their primary objective? Please explane
  • 3.    Cervical elastrography is not well explained in the introduction. For clarification, please explaine
  • 4.     The paper is well written and easily understood, but I feel that it would benefit from a review by a native English speaker

Author Response

Response to Reviewer 3 Comments

Point 1: Please, shorten the discussion

Response 1: As per the reviewer’s recommendation, we revised and shortened the discussion section.

Point 2: Why did the authors select PTD < 32 weeks gestation as their primary objective? Please explane

Response 2: We have described the rationale behind choosing PTD <32 weeks as the primary objective in the discussion of this article. Analyses concerning the prediction of PTD at <34 and <37 weeks were also conducted. There were significant results in these gestational age groups. However, compared to late preterm birth over 34 weeks, studies reported that preterm birth at <32 weeks had a greater impact on the incidence of complications related with prematurity. Since the incidence of long-term complications increases in infants who delivered before 32 weeks of gestation, early prediction and efforts to prevent premature birth are important for improving neonatal outcomes in consideration of social and medical costs. Therefore, we used 32 weeks as the PTD threshold.

Point 3: Cervical elastrography is not well explained in the introduction. For clarification, please explaine

Response 3: Thank you for your comments. We added an explanation about cervical elastography in the introduction as follows:

“Cervical elastography assesses changes in consistency during the course of the maturation process prior to delivery, such as ripening, effacement, and dilatation. Cervical elastography provides information about cervical stiffness and helps predict preterm delivery or successful labor induction [17].”

“Most studies on cervical elastography have evaluated the association between short cervix, regardless of receiving progesterone, and preterm birth [9,10,18]. These studies demonstrated that in pregnant women with short CL, the PPV for predicting preterm birth can be enhanced by adding cervical stiffness rather than using CL alone. However, its feasibility for predicting sPTD in the population with short cervix receiving progesterone treatment is unclear. The theoretical mechanism of progesterone is hypothesized to be through the inhibition of cervical ripening by controlling cervical ECM degradation and remodeling via anti-inflammatory activity and suppression of matrix metalloproteinase inhibitors [19-22]. Thus, progesterone-driven cervical changes may be detected by measuring cervical stiffness in women with short cervix who respond to progesterone treatment..”

Point 4: The paper is well written and easily understood, but I feel that it would benefit from a review by a native English speaker

Point 4:  As the reviewers suggested, we submitted the manuscript for further proofreading. Thank you for your patience.
